# Waldenström’s Macroglobulinemia in a Normoproteinemic Dog with Atypical Bimorphic Plasmacytoid Differentiation and Monoclonal Gammopathy

**DOI:** 10.3390/vetsci10050355

**Published:** 2023-05-16

**Authors:** Maud Guerlin, Kévin Mourou, Valeria Martini, Nicolas Soetart, Stefano Comazzi, Catherine Trumel, Fanny Granat

**Affiliations:** 1Département des Sciences Cliniques des Animaux de Compagnie et de Sport, Université de Toulouse, ENVT, 31000 Toulouse, France; 2CREFRE, Université de Toulouse, INSERM, ENVT, UPS, 31000 Toulouse, France; 3Département des Animaux de Compagnie, Université de Lyon, VetAgro Sup, Marcy L’Etoile, 69007 Lyon, France; 4Department of Veterinary Medicine and Animal Sciences, University of Milan, Via dell’Università 6, 26900 Lodi, Italy; 5LabOniris—Department of Biology, Pathology and Food Science, Oniris—Nantes Atlantic National College of Veterinary Medicine, Food Science and Engineering, 44300 Nantes, France; 6CRCT, Université de Toulouse, INSERM, UMR 1037, ENVT, 31000 Toulouse, France

**Keywords:** canine, electrophoresis, IgM, immunofixation, lymphoplasmacytic lymphoma, Waldenström’s macroglobulinemia, lymphocytosis, paraproteinemia

## Abstract

**Simple Summary:**

Waldenström’s macroglobulinemia is a lymphoproliferative disorder rarely described in veterinary medicine. According to the human classification of hematopoietic tumors, this disease is defined as a subtype of lymphoplasmacytic lymphoma with bone marrow involvement and monoclonal gammopathy of IgM type. Cytopathology is a convenient and accurate diagnostic procedure to diagnose lymphoma in dogs. However, ancillary techniques, such as PCR for antigen receptor rearrangement, flow cytometry, immunohistochemistry, and histology, are usually required to fully characterize the grade and phenotype, especially in ambiguous cases. Immunosecretory lymphoproliferative disorders result from malignant clonal expansion of B-cells and are usually characterized by hyperglobulinemia, although normoglobulinemia can also occur, hypoglobulinemia being rarer. Monoclonal gammopathy can be suspected on serum protein electrophoresis as it usually migrates as a narrow peak in the γ or β fractions, but immunofixation is required for confirmation and characterization. This case report describes clinicopathologic features, diagnosis, treatment, and the outcome of Waldenström’s macroglobulinemia in a 2-year-old neutered female Small Munsterlander dog with atypical bimorphic plasmacytoid differentiation and unusual M-protein migration on serum protein electrophoresis.

**Abstract:**

A 2-year-old neutered female Small Munsterlander dog was presented for an insect bite. Physical examination revealed a poor body condition, a peripheral lymphadenomegaly, and suspected splenomegaly. A complete blood count (Sysmex XN-V) revealed marked leukocytosis with lymphocytosis and abnormal dot plots. An abnormal monomorphic lymphoid population and marked rouleaux formation were noted on the blood smear. Lymph node aspirates contained an atypical bimorphic population of lymphocytes, either with a plasmacytoid or a blastic appearance. This double population was also found in the spleen, liver, bone marrow, tonsils, and other tissues. Peripheral blood and lymph node clonality assays revealed clonal BCR gene rearrangement. Flow cytometry revealed a mixed population of small-sized B-cells (CD79a+ CD21+ MHCII+) and medium-sized B-cells (CD79a+ CD21− MHCII−) in lymph nodes and a dominant population of small-sized mature B-cells (CD21+ MHCII+) in peripheral blood. Though normoproteinemic, serum protein electrophoresis revealed an increased α2-globulin fraction with an atypical restricted peak, identified as monoclonal IgM by immunofixation. Urine protein immunofixation revealed a Bence-Jones proteinuria. A diagnosis of Waldenström’s macroglobulinemia was made. Chemotherapy was initiated, but the dog was euthanized 12 months after the initial presentation due to marked clinical degradation.

## 1. Introduction

Lymphoma is a very common disease in dogs. According to the current World Health Organization (WHO), this heterogeneous group of hemolymphatic system conditions results from the neoplastic transformation of lymphoid B-cells and T-cells at different stages of development and encompasses different subtypes with varying malignancy [1,2,3]. Among B-cell lymphomas, Waldenström’s macroglobulinemia (WM) is defined as a subset of lymphoplasmacytic lymphoma with bone marrow involvement and an IgM monoclonal gammopathy (macroglobulinemia). WM is uncommon in human and veterinary medicine and is considered an indolent disease [1,2]. Clinical signs of WM are mainly related to increased serum IgM, including hyperviscosity syndrome (HVS), reported in 30% of dogs, and immune-mediated disorders. Retinal hemorrhage, epistaxis, cardiac and renal damage, and central nervous system abnormalities can be observed secondary to HVS. IgM-mediated immune disorders can manifest as cryoglobulinemia, cold agglutinin hemolytic anemia, and peripheral neuropathies [1,2]. Most common clinicopathological abnormalities in both humans and dogs consist of multifactorial anemia, pancytopenia, leukocytosis or leukopenia, thrombocytopenia, hypercalcemia, marked hyperproteinemia with IgM paraproteinemia, and Bence-Jones proteinuria [1,2,4]. M-protein can be suspected on serum protein electrophoresis as it usually migrates as a narrow peak in the γ or β fractions of the electrophoretogram, but confirmation must be done through immunofixation (IF) [1,5,6].

As only a few cases of WM are reported in veterinary medicine, there are neither well-established prognostic factors nor consensus on treatment. Different chemotherapy protocols are reported in canine WM, but they are not curative, and the treatment strategy is to reduce tumor burden and clinical signs [1]. In human patients, some factors are associated with a poor prognosis, including older age, peripheral cytopenias, especially anemia, hypoalbuminemia, genetic mutations, and increased numbers of transformed cells [2,4]. M-protein concentration does not seem to have prognostic value [4]. Although uncommon in human patients, transformation to diffuse large B-cell lymphoma occurs and is associated with poor survival [2]. 

Here, according to the WHO diagnostic criteria, we describe an unusual case of Waldenström’s macroglobulinemia (WM) in a normoproteinemic dog with atypical plasmacytoid bimorphic cell proliferation and monoclonal gammopathy [2].

## 2. Case Presentation

A 2-year-old neutered female Small Munsterlander dog was presented to the CHUVAC emergency care unit (National Veterinary School of Toulouse, France) for an insect bite on the lips. The dog had recently spent five days in a boarding facility, during which anorexia, diarrhea, and one episode of vomiting were reported. The owners also reported recent weight loss. Physical examination revealed a poor body condition (body condition score 3/9), dehydration, and generalized lymphadenomegaly. Splenomegaly and intra-abdominal lymphadenomegaly were also suspected on abdominal palpation.

Initial routine urinalysis, including specific gravity, dipstick, and direct and stained sediment examination, was unremarkable. Plasma biochemistry and electrolyte assessment (Vitros XT 3400; Ortho-Clinical Diagnostics) revealed a moderate increase of ALT (357 U/L, RI 3–50 U/L) and AST (154 U/L, RI 1–37 U/L) activities. A mildly elevated C-reactive protein (CRP, 22.5 mg/L, RI 0.5–10.0 mg/L) was also noted (CUBE-VET; Eurolyser Diagnostica GmbH). Coagulation tests (STA Compact Max2, FDP PLASMA test, Stago) revealed only a mild increase of the Fibrin Degradation Products (>5 and <20 mg/L, RI < 5 mg/L). Complete blood count (CBC, Sysmex XN-V Hematology analyzer; Sysmex Corporation) revealed a marked leukocytosis due to a marked lymphocytosis and a mild monocytosis (Table 1). On the white blood cell differential scattergram (WBC-WDF), the lymphoid population was visually predominant. A large and extended population was observed at the lymphocyte and monocyte positions and mainly identified as lymphocytes and, to a lesser extent, as monocytes, with an arbitrary separation between those two populations (Figure 1). Neutrophil and eosinophil dot plots were unremarkable. A few dots were observed in the upper right area of the WBC count scattergram (WNR).

A blood smear was stained with a modified May–Grünwald–Giemsa stain (Aerospray, Elitech Group), and a manual WBC differential count was performed by one of the authors (MG). Manual and automated WBC differential counts were in agreement. Marked rouleaux and a predominant monomorphic population of medium-sized lymphocytes were observed, with occasional increased cytoplasmic basophilia and small clear cytoplasmic inclusions (Figure 2). Numerous naked nuclei, occasional reactive lymphocytes, and rare mitotic figures were also noted. Apart from rouleaux, no significant change was observed in the erythrocyte and platelet populations. Based on CBC and blood smear examination, a lymphoproliferative disorder was suspected, although an inflammatory process with a reactive lymphocytosis could not be ruled out. 

Fine needle aspiration of enlarged peripheral lymph nodes was performed. Smears were highly cellular, with well-preserved cells and a mildly hemorrhagic background. Cytologic examination revealed a dominant atypical bimorphic lymphoid population, admixed with few plasma cells, occasional classical Mott cells, and residual mature lymphocytes (Figure 3). The first population consisted of small to medium lymphocytes (nuclei 9–10 µm in diameter) with a plasmacytoid appearance, with eccentrically placed, round, and dark nuclei with highly condensed chromatin. The cytoplasm was scant, deep blue, with frequent single or occasional multiple, small well, with defined clear blue round inclusions displacing the nucleus. This population comprised approximately 40% of the total nucleated cell population. The second population consisted of medium to large lymphoid blasts (nuclei 10 to >20 µm in diameter) with round, eccentrically placed nuclei. The chromatin was coarsely stippled with multiple small rounds to ovoid discreet nucleoli. The cytoplasm was scant and pale blue with occasional round inclusions. A low number of mitotic figures were observed. This population comprised about 35% of the total nucleated cell population. Scattered inflammatory cells were mainly composed of phagocytic macrophages; a few neutrophils and a few eosinophils were also noted. Based on lymph node fine needle aspiration cytology, lymphoma with lymphoplasmacytic differentiation was suspected as the primary differential, although an atypical reactive lymphoid population could not be ruled out.

For further evaluation, some vector-borne diseases were excluded as possible infectious causes based on a negative Snap Test 4Dx Plus (IDEXX Laboratories, Westbrook, USA) and a negative quantitative serology for leishmaniosis and ehrlichiosis. PCR for antigen receptor gene rearrangement (PARR) was performed from unstained peripheral lymph node and whole blood EDTA smears (Genefast laboratory, Forlì, Italy) and revealed a clonal BCR gene rearrangement and a polyclonal TCR gene rearrangement in both locations (Figure 4). Peripheral blood and lymph node aspirates were submitted for flow cytometric analysis (University of Milan, Lodi, Italy). The lymph node was mainly composed of a mixed population of small-sized B-cells (CD79a+ CD21+ MHCII+) and medium-sized B-cells (CD79a+ CD21− MHCII−) (Figure 5). The blood sample was composed of a dominant population of small-sized mature B-cells (CD21+ MHCII+). Cytomorphology, clonality testing, and immunophenotyping were suggestive of a leukemic B-cell lymphoma with a lymphoplasmacytic appearance.

Based on the rouleaux observation, dysproteinemia was suspected, and serum protein electrophoresis (SPE) was performed despite no significant hyperproteinemia. Agarose gel SPE (Hydrasys; SEBIA, France) revealed a decreased albumin/globulin ratio (A:G ratio, 0.65 [RI > 0.8]) and a markedly increased α_2_ globulin fraction with an atypical restricted peak suggestive of M-protein (27.7 g/L [RI 6.0–13.0]), while albumin and other globulin fractions were unremarkable (Figure 6a). Immunotyping by immunofixation (IF) was performed as described by Harris et al. and revealed a monoclonal IgM gammopathy with λ light chains (LabOniris, Nantes, France) (Figure 6b) [8]. Given SPE results, a urine protein-to-creatinine ratio was performed and was found to be increased (2.4 > 0.5). Urine protein electrophoresis (UPE, SDS-AGE, Hydrasys; SEBIA France) and immunofixation (LabOniris, Nantes, France) suggested the presence of λ free light chains indicative of Bence-Jones proteinuria (Figure 7).

Abdominal ultrasound revealed splenomegaly with multiple hypoechoic nodules, mild hepatomegaly, and multiple enlarged internal lymph nodes with heterogeneous and hypoechoic patterns. Cytological examination of splenic and hepatic fine needle aspirations revealed infiltration by the same previously reported atypical bimorphic lymphoid population, with mild to moderate hepatocellular cytoplasmic rarefaction, which could explain the increased ALT and AST activities. Bone marrow aspiration was also performed and revealed mildly increased lymphocytes and plasma cells, with infiltration by the same previously reported atypical bimorphic lymphoid population (with 2.4% of small lymphocytes with atypical cytoplasmic inclusions on manual 500 cell count and 8.4% of medium to large blastic lymphocytes) and flow cytometric results similar to the ones obtained in the peripheral blood.

### Follow-Up

Based on cytomorphology, clonality testing, and flow cytometry, the interpretation was leukemic B-cell lymphoproliferative disease with plasmacytoid appearance. Primary differentials were B-cell lymphoma with Mott-cell differentiation (MCDL), lymphoma with plasmacytoid differentiation, myeloma-related disorder, and B-cell chronic lymphocytic leukemia (B-CLL). The plasmacytoid appearance and the infiltration of the liver, spleen, and bone marrow with monoclonal IgM gammopathy led to a final diagnosis of Waldenström’s macroglobulinemia (WM) with concurrent Bence-Jones proteinuria. 

For personal reasons and because the apparent condition of the dog was stable, the owners initially declined treatment, and regular physical examinations with follow-up laboratory tests were performed. Three months after the initial presentation, cutaneous and mucosal infiltration was observed on nasopharyngeal, gingival, tonsil, and clitoral cytological examination, together with a moderate decline in general condition. Multidrug chemotherapy was initiated using L-Asparaginase (400 UI/kg, IM), followed by two CHOP-based cycles using vincristine (0.7 mg/m^2^, IV), cyclophosphamide (250 mg/m^2^, PO), doxorubicin (30 mg/m^2^ IV), and prednisolone (1 mg/kg, PO, q24 h). Improvement in general condition and clinicopathological parameters were observed. CHOP protocol was discontinued, and melphalan metronomic chemotherapy was introduced (7 mg/m^2^, PO, q24 h). The dog remained in fairly good condition, but persistent azotemia developed. After five months of metronomic chemotherapy, a moderate deterioration of the general condition and left pelvic limb lameness were reported, with cortical lysis on radiographs and infiltration on cytological examination (external laboratory). CHOP-modified chemotherapy was reintroduced, using the protocol previously described with cyclophosphamide substituted with chlorambucil (1.4 mg/kg, PO). Transient improvement of general condition and clinicopathological manifestations were observed, with complete resolution of lameness. Unfortunately, twelve months after the initial presentation, progressive and marked deterioration of the clinical condition, persistent thrombocytopenia, azotemia, and the onset of renal lesions on abdominal ultrasound were reported. The owner elected euthanasia. No necropsy was performed.

## 3. Discussion

To the best of our knowledge, this is the first report of canine WM with an atypical bimorphic cytological presentation and a restricted peak in α_2_ globulin fraction on SPE. WM is rare in humans, dogs, and cats, with only a few case reports in the veterinary literature. According to the World Health Organization (WHO), WM is included among mature B cell neoplasms and is more precisely defined as a subtype of lymphoplasmacytic lymphoma (LPL) with bone marrow involvement and IgM monoclonal gammopathy (i.e., macroglobulinemia) at any concentration [2]. 

In the current case, concurrent observation of abnormal WBC scattergrams and marked lymphocytosis was primarily suggestive of an atypical lymphoid population. A large and extended population was observed on the WDF channel at the lymphocyte and monocyte positions with an arbitrary separation, and a few abnormal events were also observed on the WNR channel. The absence of clear separation between the different leukocyte populations and the presence of cells with higher fluorescent activity, such as reactive, atypical, or blastic lymphoid cells in the monocyte position, have been reported in dogs and people with acute leukemia or leukemic lymphoma when blood was analyzed with different Sysmex analyzers (XT-2000 iV and XN-V). The few events observed in the upper right area on the WNR scattergram have also been previously reported in canine leukemia cases with the Sysmex XT-2000 iV and, more recently, with the Sysmex XN-V. Cells in this location were designated as lysis-resistant cells [7,9]. Moreover, the abundant monomorphic population of medium-sized lymphocytes on the blood smear with occasional cytoplasmic clear inclusions further strengthened the suspicion of a lymphoproliferative disorder, although reactive expansion could not be ruled out, thus requiring further investigation, such as clonality testing and flow cytometry, to confirm our suspicion.

The atypical bimorphic population observed on cytological examination and composed of small mature lymphocytes with cytoplasmic inclusions and medium to large-sized blasts have already been reported in cases of Mott cell differentiation lymphoma (MCDL), with various appellations, such as B-cell lymphoma with plasmacytoid differentiation and atypical cytoplasmic inclusions or B-cell lymphoma with plasmacytoid and/or Mott cell differentiation [10,11,12,13,14,15,16,17,18]. MCDL is uncommon, with few case reports in dogs, cats, and ferrets, making it a poorly defined disease whose classification needs to be clarified. Among these publications, two canine cases in particular share a cytomorphology very similar to our case [10,11], whereas the other cases demonstrate intermediate to large blasts admixed with numerous Mott cells with several clear globules consistent with Russel bodies distributed in the background [12,13,14,15,16,17,18]. This latter aspect was not observed in our case. MCDL was initially described as a cytomorphologically unique form of non-secretory, primarily gastrointestinal lymphoma, with 54% of cases (6/11) involving various sites of the small intestine (duodenum, jejunum, ileum) or, less frequently, the stomach [11,14,16,17,18]. Most recent cases are multicentric lymphomas similar to our case, with nodal, splenic, hepatic, renal, or bone marrow involvement without consistent primary digestive infiltration. In the current case, no gastrointestinal lesions were observed on abdominal ultrasound. Among published cases of MCDL, the majority of cases were normoproteinemic (81%), with only two cases presenting mild hypoproteinemia (18%). M-protein is not commonly reported in MCDL but is also usually not explored, with only one study performing IF. Only two cases of MCDL with confirmed monoclonal gammopathies are reported, based on IF and immunoglobulin quantification, including one of IgA type and another of IgM type [12]. 

It is noteworthy that the lymphoid population in blood was cytomorphologically different from the bimorphic population observed in solid lesions, which have already been reported in one previous case of MCDL [10]. Thus, ancillary tests were performed to confirm leukemic involvement. Flow cytometry revealed two different B-cell populations in the lymph node aspirate, both staining positive for CD79a, but with the more common one staining negative for CD21 and the other staining positive. The presence of two populations was compatible with the bimorphic cytological picture, although it was not possible to definitively prove which morphological subtype corresponded to which immunophenotype. In the peripheral blood, more than 80% of the circulating lymphoid cells shared the same immunophenotype, thus supporting their neoplastic nature [1], and labeled positive for CD21 and MHCII, as did one of the two populations observed in the nodal sample. This phenotype has already been reported in a previous case of MCDL, which shared a very similar cytomorphology to our case [10]. PARR confirmed the presence of a clonal lymphoid population in both the lymph node and the peripheral blood. Unfortunately, we did not have the opportunity to sort cells and perform PARR on the two discrete nodal populations. Thus, the reactive nature of the plasmacytoid population could not be definitively excluded, although this is very unlikely with a B-cell lymphoma.

Normoproteinemia and the serum protein electrophoretogram were also unusual features of our case. Given rouleaux, lymphocytosis, nodal atypical cytomorphology, and decreased A:G ratio, dysproteinemia was suspected, though the dog was normoproteinemic. SPE revealed an atypical restricted peak in the α_2_ globulin fraction, confirmed by monitoring SPE at several weeks intervals and by inter-laboratory testing (Laboratoire Central de l’ENVT, LabOniris). IF then identified M-proteins as a complete IgM with λ light chains. M-proteins usually migrate in γ or β fractions in SPE tracing [1,19]. Increased α_2_ globulin fraction can be observed in acute phase response (e.g., haptoglobin, ceruloplasmin), nephrotic syndrome (i.e., α_2_-lipoprotein, α_2_-macroglobulin), or with an interfering substance (e.g., lipemia) [1]. In canine cases of WM, marked hyperproteinemia is usually observed, with IgM migration in the γ or β fractions [20,21,22,23,24,25]. Rare cases of M-protein migration in α_2_ fraction have been reported in human multiple myeloma and monoclonal gammopathy of undetermined significance (MGUS), with mostly IgA and less commonly IgG or free light chains [26,27]. Several hypotheses have been proposed to explain this atypical migration, including complex formations with other plasma components, high carbohydrate content, or IgA properties, such as polymerization. In two previous cases of MCDL, dogs were normoproteinemic with variably discrete peaks observed in the α_2_ fraction on SPE [12]. Peaks were primarily attributed to acute phase response, while IF revealed the presence of M-proteins in both cases and, more precisely, monoclonal IgA and IgM, respectively. The authors then suspected that α_2_ peaks were, in fact, due to M-proteins and possibly masked by acute phase proteins (APP) for one case. One case of monoclonal IgG(T) migration in the α_2_ fraction has also been reported in a case of multiple myeloma in a mare [28]. Our current report emphasizes the importance of immunofixation for detecting monoclonal gammopathy in normoproteinemic suspect cases and when M-protein is present in low concentration or possibly masked by APPs, as recommended by recent publications [29,30]. 

The present case fulfills the criteria for WM based on cytomorphology, immunophenotyping, IgM paraproteinemia, and WHO classification, though other IgM-secreting B-cell neoplasms with plasmacytoid differentiation could be considered as potential differential diagnoses [2,4]. Unfortunately, histology and immunochemistry performed on stained slides or on cell blocks, including labeling for MUM-1, Pax-5, and IgM, special stainings such as periodic acid-Schiff (PAS), or ultrastructural microscopy have not been performed to further characterize this atypical bimorphic population, as previously reported [10,11,12,13,14,15,16,17,18]. 

Clinical signs of WM include hyperviscosity syndrome, cryoglobulinemia, bleeding tendency, and renal damage, and are mainly related to increased serum IgM content [1,25]. In our case, only a few clinicopathological signs were observed, including Bence-Jones proteinuria and, later, azotemia and thrombocytopenia, and were probably due to the subnormal proteinemia and the mild paraproteinemia. The unremarkable urinalysis with a normal dipstick at initial presentation could be due to its relative insensitivity to paraproteinuria. The dog developed renal azotemia 7 months after initial presentation, probably secondary to persistent proteinuria, IgM deposits, neoplastic infiltration, and possibly, hyperviscosity syndrome. Persistent thrombocytopenia occurred 12 months after initial presentation, possibly secondary to doxorubicin chemotherapy or multifactorial mechanisms (immune-mediated, neoplastic). 

WM is considered an indolent disease. Treatment is not curative, and various combinations of prednisone and alkylating agents can be used to reduce tumor burden and clinical signs, but prognostic factors for treatment response and survival are lacking [1]. Approximately 77% of dogs respond favorably to chemotherapy, although different biological behaviors are reported in the veterinary literature, from a hyperacute aggressive clinical course to durable remission over 42 months [1,21,22,23,25]. In our case, the initial CHOP chemotherapy showed primarily good results based on the patient’s condition and clinicopathological assessment, including durable reduction of lymphocytosis and clitoromegaly. In human medicine, the transformation of WM into diffuse large B-cell lymphoma (DLBCL) occurs in a small proportion of cases and is associated with poor survival outcomes [2]. Whether aggressive transformation can also occur in canine WM remains unknown, although it could be suspected from the hyperacute aggressive behavior and variable response to therapy previously reported in dogs. In the current case, disease progression occurred after 5 months of melphalan metronomic chemotherapy. Ultimately, marked deterioration occurred one year after the initial presentation despite CHOP reinduction. Transformation to a high-grade lymphoma could not be ruled out since no final cytological and histological examination was performed. 

## 4. Conclusions

This case report describes a rare case of canine Waldenström’s macroglobulinemia with two unusual features. First, the dog was almost normoproteinemic, with only a moderate decrease in the albumin:globulin ratio. However, marked rouleaux observed on the blood smear and the atypical restricted peak in the α_2_ globulin fraction of the SPE tracing led to a suspicion of dysproteinemia, which was further characterized as monoclonal IgM. This finding emphasizes the crucial role of accessible tests, such as SPE, and more advanced analysis techniques, notably IF and immunoglobulin quantification, in order to detect monoclonal gammopathy in normoproteinemic cases or suspicious SPE tracing. Second, the bimorphic population with plasmacytoid differentiation was atypical. This cytological presentation has been reported previously in rare cases of MCDL but, to the best of our knowledge, has never been reported in canine WM. The cytomorphologic discrepancy between neoplastic cells in peripheral blood and those in the solid lesions emphasizes the importance of ancillary diagnostic techniques, including immunophenotyping and clonality testing, for more accurate tumor staging in dogs with lympho-proliferative disorders.

## Figures and Tables

**Figure 1 vetsci-10-00355-f001:**
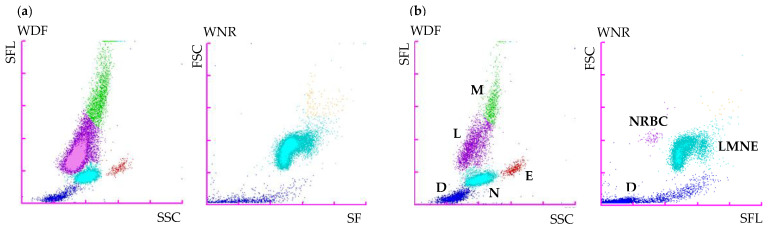
(**a**) Sysmex XN-V WBC scattergrams from a Small Munsterlander dog with WM and (**b**) a healthy dog for cell identification from a previous study [7]. Abbreviations: D, debris; E, eosinophils; FSC, forward scatter light; L, lymphocytes; M, monocytes; N, neutrophils; NRBC, nucleated red blood cells; SFL, side fluorescent light; SSC, side scatter light; WDF, WBC differential scattergrams; WNR, WBC count scattergrams.

**Figure 2 vetsci-10-00355-f002:**
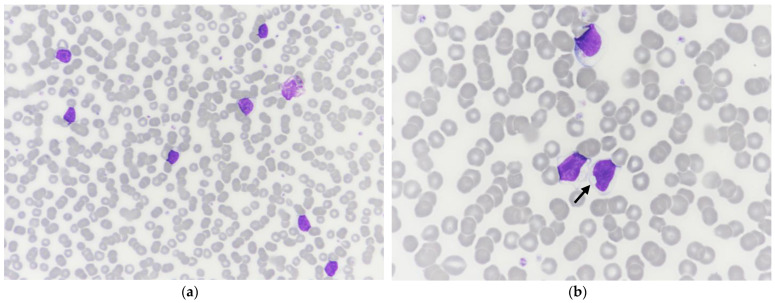
Peripheral blood smear from a Small Munsterlander dog with WM (modified May–Grünwald–Giemsa stain, (**a**): ×500; (**b**): ×1000, oil). Note the marked rouleaux formation and the homogeneous population of medium-sized mature lymphocytes (nuclei 10–14 µm in diameter) with clumped chromatin and a scant to moderate amount of pale blue cytoplasm with occasional small clear inclusions (arrow).

**Figure 3 vetsci-10-00355-f003:**
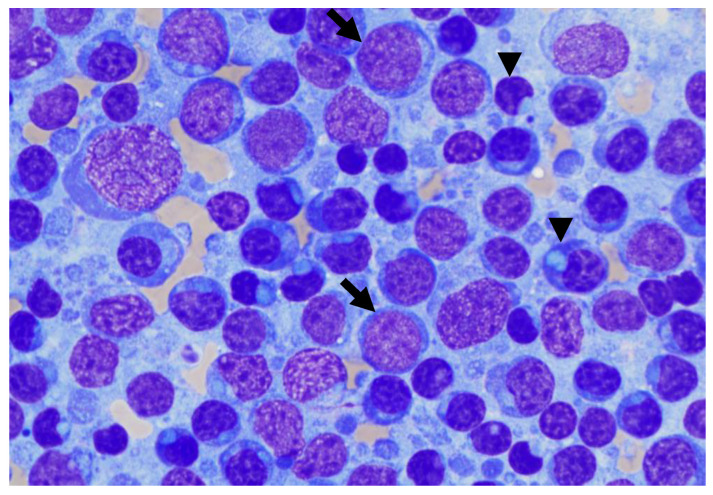
Prescapular lymph node fine needle aspiration from a Small Munsterlander dog with WM (modified May–Grünwald–Giemsa stain, ×1000, oil). Note the atypical bimorphic lymphoid population composed of small to medium lymphocytes with plasmacytoid appearance (arrow-head) and medium to large blasts (arrow), admixed with few plasma cells and residual mature lymphocytes.

**Figure 4 vetsci-10-00355-f004:**
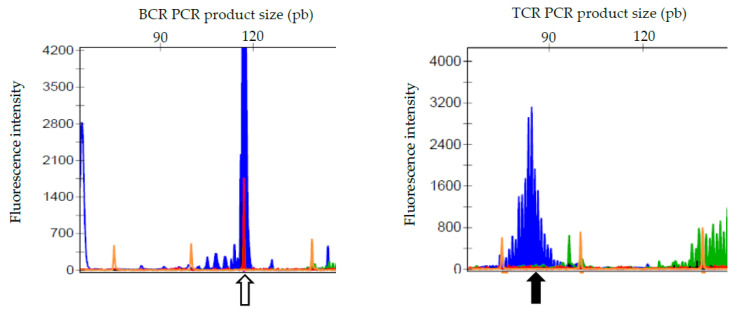
Selected PARR graphics of a Small Munsterlander dog with WM from a peripheral lymph node specimen, showing clonal BCR gene rearrangement pattern (open arrow) and polyclonal TCR gene rearrangement (solid arrow).

**Figure 5 vetsci-10-00355-f005:**
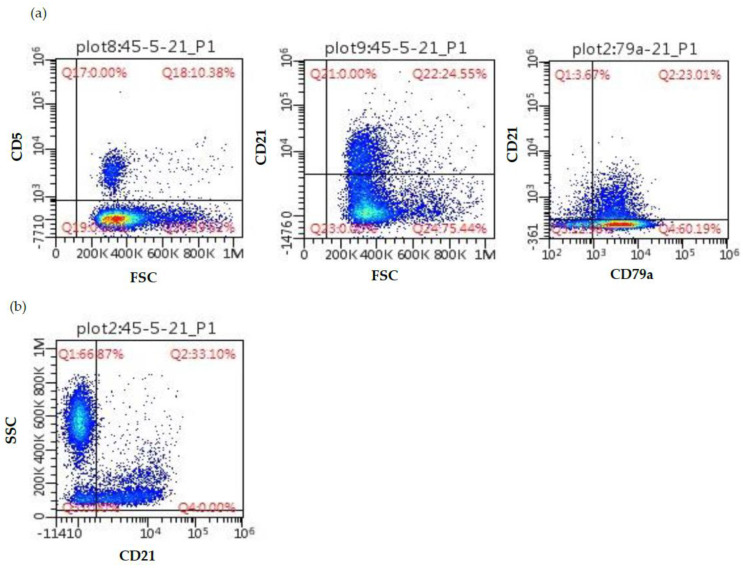
Flow cytometric scatter plots from a Small Munsterlander dog with WM. (**a**) peripheral lymph node; (**b**) peripheral blood. The lymph node sample is composed of a mixed population including 10% small-sized T-cells (left panel), 25% small-sized B-cells staining positive for CD79a and CD21 (central and right panel), and a third population (60%) of B-cells staining positive for CD79a but negative for CD21 (right panel). The blood sample is composed of 33% small-sized B-cells staining positive for CD21 (accounting for 82.5% of circulating lymphoid cells).

**Figure 6 vetsci-10-00355-f006:**
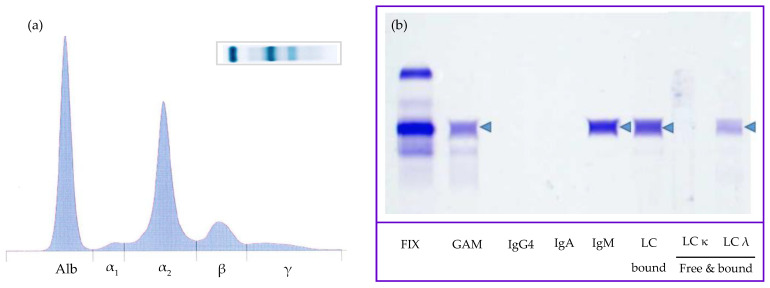
Serum protein electrophoresis (**a**) and immunofixation (IF) (**b**) from a Small Munsterlander dog with WM (Hydrasys; SEBIA, France). Proteinemia was at the upper limit of the reference interval (Total proteins: 67 g/L [reference interval: 48–66 g/L], Vitros XT 3400, Ortho-Clinical Diagnostics). SPE reveals a markedly increased α2 globulin fraction with an atypical restricted peak. IF reveals an M-Protein identified as a complete immunoglobulin with µ heavy chain (IgM) and λ light chain (blue arrowhead). Abbreviations: Alb, albumin; GAM, heavy chains polyclonal (γ (IgG), α (IgA), μ (IgM)); LC, light chain; MW, molecular weight; FIX, all protein (please find the whole serum protein electrophoresis and immunofixation gels in Appendix A and Appendix A, respectively).

**Figure 7 vetsci-10-00355-f007:**
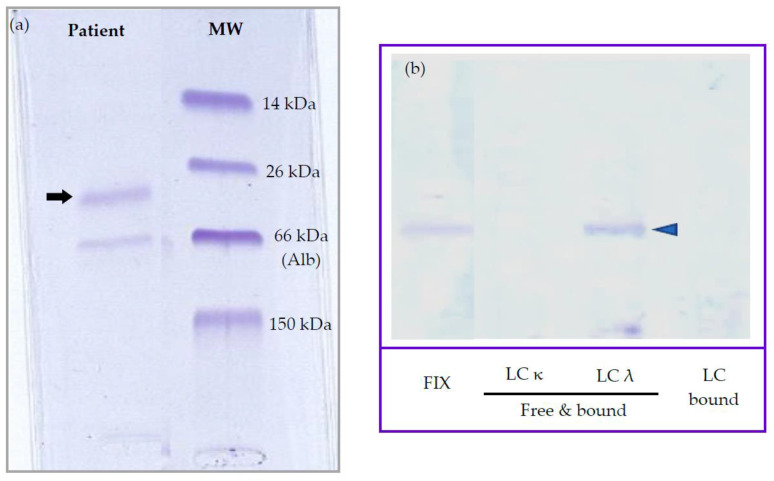
Urine protein electrophoresis (SDS-AGE) (**a**) and immunofixation (**b**) from a Small Munsterlander dog with WM. SDS-AGE shows one restricted band between 26 and 66 kDa (arrow). Urine immunofixation reveals a distinct restricted band identified as free λ light chains (blue arrow-head), consistent with Bence-Jones proteinuria. Abbreviations: LC, light chain; SDS-AGE, sodium dodecyl sulfate agarose gel electrophoresis; MW, molecular weight; FIX, all proteins. (Please find the whole urine protein electrophoresis and immunofixation gels in Appendix A and Appendix A, respectively).

**Table 1 vetsci-10-00355-t001:** Automated and manual hematologic results from a Small Munsterlander dog with Waldenström’s macroglobulinemia (WM) (Sysmex XN-V Hematology analyzer, Sysmex Corporation, Kobe, Japan). Manual PCV and differential count values are between brackets. The bolded values indicate a deviation from the reference interval.

	Unit	Result	Reference Interval
HGB	g/dL	15.5	12.4–19.2
RBC	×10^12^/L	6.72	5.20–7.90
HCT	L/L	0.45 (0.45)	0.35–0.52
MCV	fL	67.0	60.0–71.0
MCH	pg	23.1	21.9–26.3
MCHC	g/dL	34.4	34.4–38.1
RDW-SD	fL	**30.7**	31.1–38.9
RDW-CV	%	13.2	13.2–19.1
PLT-I	×10^9^/L	181	64–613
PLT-O	×10^9^/L	195	108–562
WBC-WNR	×10^9^/L	**34.37**	5.60–20.40
Neutrophils	×10^9^/L	7.82 (5.84)	2.90–13.60
Lymphocytes	×10^9^/L	**24.53 (26.12)**	1.10–5.30
Monocytes	×10^9^/L	**1.72 (1.72)**	0.40–1.60
Eosinophils	×10^9^/L	0.19 (0.34)	<3.10
Reticulocytes	×10^9^/L	10.10	19.40–150.10
Reticulocytes	%	0.15	0.30–2.37

Abbreviations: HCT, hematocrit; HGB, hemoglobin concentration; MCH, mean corpuscular hemoglobin; MCHC, mean corpuscular hemoglobin concentration; MCV, mean corpuscular volume; PCV, packed cell volume; PLT-I and PLT-O, impedance and optical platelet counts, respectively; RBC, impedance RBC count; RDW-SD and RDW-CV, red cell distribution width standard deviation and coefficient of variation, respectively; WBC-D and WBC-WNR, WBC counted on the differential channel and the white cell nucleated channels, respectively.

## Data Availability

Not applicable.

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
