# Peer review of "Waldenström’s Macroglobulinemia in a Normoproteinemic Dog with Atypical Bimorphic Plasmacytoid Differentiation and Monoclonal Gammopathy"

_vetsci, 2023, doi:10.3390/vetsci10050355_

Round 1

Reviewer 1 Report

In the case report the authors describe a case of a B-cell lymphoid malignancy associated with an IgM paraprotein and uncommon cytologic features. The clinical case is well characterized in terms of hematologic, cytologic, flow cytometry and protein electrophoresis and immunofixation results. The follow up was also included and detailed. The major flaw of the case is the absence of the gold standard test that is the histologic examination (of lymph node biopsy at the time of the diagnosis or biopsies obtained post-mortem) that would allow the definitive diagnosis of the type of the lymphoid malignancy and confirmed the extension to liver, spleen and bone marrow. This also jeopardized the determination of the cause of death (did the disease progress to a high-grade lymphoma or not?).

Moreover, the authors stated in the discussion (lines 390-393) that immunolabelling or special stains (PAS) of the lymphoid cells was not possible. I assume that the authors said that because no tissue biopsy was obtained, however it should be recall that there is evidence that lymphoid markers, such as CD3, PAX5, CD20, MUM1 can be detected in previously stained cytology smears. Additionally, special stains, such as PAS also work very well in previous stained cytology specimens. The inclusion of the immunocytochemistry as an ancillary technique (by using B cell markers on the previous stained/archived slides of the FNA of lymph node, spleen, liver and bone marrow) would certainly increase the value of the report and would allow the correlation of the flow cytometry and PARR results with a morphologic demonstration of the B cell markers in the neoplastic lymphocytes.

Minor points

Simple summary: the term immunohistopathology should be replace by immunohistochemistry and histology

Abstract: were tonsils sampled during the diagnostic workup? If not, this should be deleted from the abstract.

Case presentation

lines 263-264: Which was the count of the atypical biomorphic lymphocytes in bone marrow (the % of blasts was not mentioned and should be included)

line 267: rather than “outcome” the subtitled should be “follow-up”

Discussion:

Lines 390-393: this should be reformulated according to the comment above. The authors should also discuss the utility of cell blocks obtained from needle rinses of lymph nodes (specially in cases like this, for which the owner did not allow surgical biopsy and necropsy).

Lines 400-401: the sentence should be reformulated so that it would not start with “7 months”

Author Response

In the case report the authors describe a case of a B-cell lymphoid malignancy associated with an IgM paraprotein and uncommon cytologic features. The clinical case is well characterized in terms of hematologic, cytologic, flow cytometry and protein electrophoresis and immunofixation results. The follow up was also included and detailed. The major flaw of the case is the absence of the gold standard test that is the histologic examination (of lymph node biopsy at the time of the diagnosis or biopsies obtained post-mortem) that would allow the definitive diagnosis of the type of the lymphoid malignancy and confirmed the extension to liver, spleen and bone marrow. This also jeopardized the determination of the cause of death (did the disease progress to a high-grade lymphoma or not?).

Nasal biopsies were performed two months after her presentation to the veterinary school and before the initiation of chemotherapy and concluded to a lymphomatous infiltration of the nasal mucosa. No immunological diagnosis had been made on these samples.

It is true that the exact cause of death could not be determined as there was no autopsy. Only hypotheses can be made on the basis of the latest clinical and biological results.

Moreover, the authors stated in the discussion (lines 390-393) that immunolabelling or special stains (PAS) of the lymphoid cells was not possible. I assume that the authors said that because no tissue biopsy was obtained, however it should be recall that there is evidence that lymphoid markers, such as CD3, PAX5, CD20, MUM1 can be detected in previously stained cytology smears. Additionally, special stains, such as PAS also work very well in previous stained cytology specimens. The inclusion of the immunocytochemistry as an ancillary technique (by using B cell markers on the previous stained/archived slides of the FNA of lymph node, spleen, liver and bone marrow) would certainly increase the value of the report and would allow the correlation of the flow cytometry and PARR results with a morphologic demonstration of the B cell markers in the neoplastic lymphocytes

We thank the reviewer for these remarks and agree that immunocytochemistry would have been interesting and would have allowed to confirm our diagnosis. We also agree that it could have been done on the already stained slides. However, it seemed to us that the whole of the examinations carried out were sufficient evidence. Moreover, the time for the corrections being limited to 5 days, it is impossible for us to carry out these additional experiments in the time allotted. Finally, for budgetary considerations, it was difficult for us to carry out all these additional markings.

Minor points

Simple summary: the term immunohistopathology should be replace by immunohistochemistry and histology

Correction done.

Abstract: were tonsils sampled during the diagnostic workup? If not, this should be deleted from the abstract.

Tonsil were sampled during the diagnostic workup.

Case presentation:

lines 263-264: Which was the count of the atypical biomorphic lymphocytes in bone marrow (the % of blasts was not mentioned and should be included)

Correction done : “Bone marrow aspiration was also performed and revealed mildly increased lymphocytes and plasma cells, with an infiltration by the same previously reported atypical bimorphic lymphoid population (with 2.4% of small lymphocytes with atypical cytoplasmic inclusions on manual 500 cell count and 8.4 % of medium to large blastic lymphocytes)”

line 267: rather than “outcome” the subtitled should be “follow-up”

Correction done

Discussion:

Lines 390-393: this should be reformulated according to the comment above. The authors should also discuss the utility of cell blocks obtained from needle rinses of lymph nodes (specially in cases like this, for which the owner did not allow surgical biopsy and necropsy).

Correction done : ”Unfortunately, histology and immunochemistry performed on stained slides or on cell blocks including labelling for MUM-1, Pax-5, and IgM, special staining such as periodic acid-Schiff (PAS), or ultrastructural microscopy have not been performed to further characterize this atypical bimorphic population, as previously reported”.

Lines 400-401: the sentence should be reformulated so that it would not start with “7 months”

Correction done : ”The dog developed renal azotemia 7 months after initial presentation, probably secondary to persistent proteinuria, IgM deposits, neoplastic infiltration and possibly hyperviscosity syndrome”.

Reviewer 2 Report

Dear authors,

This article is very well written and describes a rare case of Waldenström disease in a young dog. This atypical condition has been well documented, by cross-referencing several techniques and repeating the examinations in different laboratories for electrophoresis. The description of changes in the blood cell analyser dot clouds (an abnormal cloud in the monocytes place) prompted cytological analyses of the blood smear (red blood cell rolls and atypical population of small monomorphic lymphocytes), and led to the suspicion of dysprotidemia and lymphocyte abnormality. Fine needle aspiration of the organs modified on clinical examination and ultrasound (Lymph nodes, spleen, liver), and of the bone marrow made it possible to describe, apart from the blood and the bone marrow, an infiltration of the other haematopoietic organs by a population of lymphocytes of plasmacytic morphology characterised with flow cytometry and with the PARR technique in B lymphocytes (B receptor clonal rearrangement). Serum protein electrophoresis showed a monoclonal peak in alpha 2 in the absence of hyperproteinemia, and immunofixation of these proteins classified them as Immuglobulin M (mu heavy chain and lambda light chain).

Despite the initial absence of evidence of proteinuria on routine examination of the urine, a proteinuria/creatinine ratio was found to be above the threshold, electrophoresis of the urine proteins was then carried out, and their identification by immunostaining enabled the identification of gamma light chain corresponding to a Bence Jones proteinuria. The authors must be commended for their tenacity in clarifying this case, for which nothing was easy or obvious to guide the diagnosis, and for the rigour of their diagnostic approach, which made it possible to support Waldenström's disease with numerous cross-disciplinary techniques. The only point that could be deplored about the diagnostic steps is the absence of histological sampling of the lymph nodes, which would have made it possible, with different techniques, in particular immunolabelling, to better characterize each of the 2 atypical populations of B lymphocytes described (small cell, and medium to large cell): the PARR technique having shown a clonal rearrangement of the B receptor on blood and on lymph nodes, it would have been interesting to know if these two cell populations actually presented the same clonal rearrangement of their receptor and if this bimorphic population could have corresponded from the outset to an escape in high grade lymphoma, having been able to respond to the chemotherapy treatment initiated at the outset, and which could explain the aggressive evolution of the disease later on. Or, if the second population was just a reactive plasmocytoid population.

The main focus of this article is the diagnosis of this rare and very atypical disease. The follow-up of the dog is described less precisely, and one can regret the absence of follow-up (or its description) of serum and urine protein electrophoresis, or even the absence of follow-up of the proteinuria/creatinuria ratio; did the monoclonal peak disappear as expected under CHOP chemotherapy? then under melphalan? The renal disease could perhaps have been delayed with a treatment aimed at protecting the kidneys (diet, sartans, etc.). The relapse on the other hand (lameness with bone invasion by the tumour) is well described cytologically as a relapse of this clonal lymphoproliferative disorder with plasmocytoid differentiation.

Congratulations on this very nice clinical case!

Then just a few details:

line 91: Could you approximate the sum of the diameters of the main LN? the poor general condition is contradicted in line 275

line 161: on blood smear you describe medium sized lymphocytes (in abstract line 43-44 and 172 you wrote small lymphocytes.

line 203, I suppose serological tests for ehrlichia and leishmania were unremarkable? (please detail the results)

Author Response

This article is very well written and describes a rare case of Waldenström disease in a young dog. This atypical condition has been well documented, by cross-referencing several techniques and repeating the examinations in different laboratories for electrophoresis. The description of changes in the blood cell analyser dot clouds (an abnormal cloud in the monocytes place) prompted cytological analyses of the blood smear (red blood cell rolls and atypical population of small monomorphic lymphocytes), and led to the suspicion of dysprotidemia and lymphocyte abnormality. Fine needle aspiration of the organs modified on clinical examination and ultrasound (Lymph nodes, spleen, liver), and of the bone marrow made it possible to describe, apart from the blood and the bone marrow, an infiltration of the other haematopoietic organs by a population of lymphocytes of plasmacytic morphology characterised with flow cytometry and with the PARR technique in B lymphocytes (B receptor clonal rearrangement). Serum protein electrophoresis showed a monoclonal peak in alpha 2 in the absence of hyperproteinemia, and immunofixation of these proteins classified them as Immuglobulin M (mu heavy chain and lambda light chain).

Despite the initial absence of evidence of proteinuria on routine examination of the urine, a proteinuria/creatinine ratio was found to be above the threshold, electrophoresis of the urine proteins was then carried out, and their identification by immunostaining enabled the identification of gamma light chain corresponding to a Bence Jones proteinuria. The authors must be commended for their tenacity in clarifying this case, for which nothing was easy or obvious to guide the diagnosis, and for the rigour of their diagnostic approach, which made it possible to support Waldenström's disease with numerous cross-disciplinary techniques. The only point that could be deplored about the diagnostic steps is the absence of histological sampling of the lymph nodes, which would have made it possible, with different techniques, in particular immunolabelling, to better characterize each of the 2 atypical populations of B lymphocytes described (small cell, and medium to large cell): the PARR technique having shown a clonal rearrangement of the B receptor on blood and on lymph nodes, it would have been interesting to know if these two cell populations actually presented the same clonal rearrangement of their receptor and if this bimorphic population could have corresponded from the outset to an escape in high grade lymphoma, having been able to respond to the chemotherapy treatment initiated at the outset, and which could explain the aggressive evolution of the disease later on. Or, if the second population was just a reactive plasmocytoid population.

We sincerely thank the reviewer for his comments and remarks. We strongly agree that a histopathologic examination of a lymph node with additional immunostaining would have been very interesting. Histopathologic examination of nasal cavity biopsy specimens was also performed and confirmed lymphoma. However, for financial reasons, immunohistochemical examinations were not performed at first because they were refused by the owner. Finally, as all the results were sufficiently conclusive, we decided not to carry out these immunomarkings, notably once again for financial reasons.

The main focus of this article is the diagnosis of this rare and very atypical disease. The follow-up of the dog is described less precisely, and one can regret the absence of follow-up (or its description) of serum and urine protein electrophoresis, or even the absence of follow-up of the proteinuria/creatinuria ratio; did the monoclonal peak disappear as expected under CHOP chemotherapy? then under melphalan? The renal disease could perhaps have been delayed with a treatment aimed at protecting the kidneys (diet, sartans, etc.). The relapse on the other hand (lameness with bone invasion by the tumour) is well described cytologically as a relapse of this clonal lymphoproliferative disorder with plasmocytoid differentiation.

We agree that the lack of follow-up of serum and urine protein electrophoresis is quite regrettable, especially after the initiation of chemotherapy. But for financial reasons, it was not possible to do these controls. On the other hand, RPCU could be carried out before and after the implementation of chemotherapy. Values ranging from 0.918 to 2.43 could be observed before chemotherapy and a value of 0.16 was measured after the implementation of melphalan.

few details:

- line 91: Could you approximate the sum of the diameters of the main LN?

The lymph nodes were unfortunately not measured. On the other hand, a CT scan of the head was performed before the chemotherapy was started. The mandibular and retropharyngeal lymph nodes were measured and evaluated at 30 mm in length.

 the poor general condition is contradicted in line 275: “dog's apparent good general condition”

Correction done : “For personal reasons and because the apparent condition of the dog was stable, the owners initially declined treatment, and regular physical examinations with follow-up laboratory testing were performed”.

line 161: on blood smear you describe medium sized lymphocytes (in abstract line 43-44 and 172 you wrote small lymphocytes.

The differences in size are related to the different techniques used. Cytometry and microscopy do not always give completely similar information and can explain this difference.

line 203, I suppose serological tests for ehrlichia and leishmania were unremarkable? (please detail the results)

Information have been added:  “For further evaluation, some vector borne diseases were excluded as possible infectious causes based on a negative Snap Test 4Dx Plus (IDEXX Laboratories, Westbrook, USA) and a negative quantitative serology for leishmaniosis and ehrlichiosis”.
